# White Matter Hyperintensities Mediate the Negative Impact of HbA1c Levels on Cognitive Function

**DOI:** 10.3390/brainsci15070692

**Published:** 2025-06-27

**Authors:** Rudolph Johnstone, Ida Rangus, Natalie Busby, Janina Wilmskoetter, Nicholas Riccardi, Sarah Newman-Norlund, Roger Newman-Norlund, Chris Rorden, Julius Fridriksson, Leonardo Bonilha

**Affiliations:** 1School of Medicine Columbia, University of South Carolina, 6311 Garners Ferry Rd, Columbia, SC 29209, USA; 2Department of Communication Sciences and Disorders, University of South Carolina, 915 Greene Street, Columbia, SC 29208, USA; 3Department of Rehabilitation Sciences, Medical University of South Carolina, 160 Ashley Avenue, Charleston, SC 29425, USA; 4Department of Psychology, University of South Carolina, 915 Greene Street, Columbia, SC 29208, USA

**Keywords:** diabetes, cognitive impairment, white matter hyperintensities, dementia, brain aging

## Abstract

Background: Type 2 diabetes is linked to impaired cognitive function, but the underlying mechanisms remain poorly understood. As white matter hyperintensities (WMHs) are common in diabetes and associated with vascular brain injury, we investigated whether WMH burden mediates the relationship between hemoglobin A1c (HbA1c) levels and cognition. Methods: We quantified WMH load using the Fazekas scale and conducted a mediation analysis with HbA1c as the independent variable, Fazekas scale as the mediator, and MoCA scores as the outcome variable. Results: WMHs partially mediated the relationship between HbA1c levels and MoCA scores (indirect effect = −0.224, 95% CI = −0.619 to −0.050, *p* = 0.001), accounting for approximately 15.6% of the total effect. Conclusions: This study suggests that WMHs partially mediate the association between chronically elevated blood glucose levels and cognitive impairment in neurologically healthy adults, supporting a potential microvascular mechanism in diabetes-related cognitive impairment.

## 1. Introduction

Hemoglobin A1c (HbA1c) is a measure of glycated hemoglobin that reflects average blood glucose levels over the previous 2 to 3 months, with levels over 6.5% being diagnostic for diabetes [1]. The relationship between elevated HbA1c, as seen in type 2 diabetes, and impaired cognitive function is well-established, though the underlying mechanisms remain under investigation [2,3]. A better understanding of these mechanisms is crucial for developing effective treatments and preventive strategies, which could reduce the burden on healthcare systems and enhance individuals’ quality of life. This need is particularly pressing given that the global prevalence of diabetes is expected to rise to 783 million people, with expected healthcare expenditures of over USD 1 trillion by 2045 [4].

Several mechanisms have been proposed to explain the link between diabetes and cognitive decline, including cerebral insulin resistance, chronic inflammation, and microangiopathy [5,6]. Chronic hyperglycemia contributes to each of these processes, ultimately causing oxidative stress and neuronal injury or apoptosis, which likely underlie the resulting cognitive dysfunction [7]. Both inflammation and microangiopathy also significantly contribute to the development of white matter hyperintensities (WMHs) [8], areas of increased signal intensity on T2-weighted MRI scans that reflect white matter and cerebral small vessel damage [9]. In type 2 diabetes, factors such as endothelial cell dysfunction, oxidative stress, advanced glycation end products, and inflammation may be directly responsible for this damage [10]. Specifically, the thickening and injury of small blood vessels can lead to decreased blood flow to critical areas of the brain, potentially resulting in the formation of WMHs and contributing to cognitive impairment [11].

Studies have established a link between both type 2 diabetes and prediabetes and an increased volume of WMHs [12,13,14]. Additionally, growing evidence identifies WMHs as a significant factor in cognitive impairment, particularly in individuals with certain metabolic conditions and cardiovascular diseases such as hypertension, carotid atherosclerosis, atrial fibrillation, and diabetes [15,16]. The damage to white matter integrity from WMHs may impair the efficiency of neural networks, ultimately resulting in cognitive deficits [17]. However, the mechanistic link between elevated hemoglobin A1c (HbA1c) levels, WMHs, and cognition has not yet been established. Mediation analysis offers a valuable framework to explore potential indirect effects [18], helping to clarify whether the relationship between elevated HbA1c levels and cognitive impairment is mediated by the presence of WMHs. By modeling these pathways, this approach provides insights into how changes in white matter integrity may serve as an intermediary mechanism linking metabolic dysfunction (as reflected by elevated HbA1c levels) to cognitive decline. A better understanding of these mechanisms could inform targeted interventions to mitigate cognitive impairment in those at risk. Therefore, our aim was to investigate whether elevated HbA1c levels contribute to cognitive impairment through the mediation of WMHs.

## 2. Materials and Methods

### 2.1. Participants

We analyzed the data of 205 neurologically intact adults. Participants were drawn from the ongoing Aging Brain Cohort (ABC) Study at the University of South Carolina (ABC@USC) [19], which aims to investigate the determinants of brain health across the lifespan by performing extensive neuroimaging and behavioral testing. The inclusion criteria required participants to be aged 20 to 80 years with no prior history of neurodegenerative disease or stroke [19]. Additionally, individuals were excluded if they had serious acute or chronic conditions that limited their ability to participate, severe current illnesses (e.g., cancer), psychiatric diagnoses (e.g., schizophrenia), or a BMI greater than 42 kg/m^2^ [19]. All participants with available MRI scans and cognitive testing via the Montreal Cognitive Assessment (MoCA) Test [20] were considered for this analysis (*n* = 205). Demographic information is presented in Table 1. This study was approved by the institutional review board at the University of South Carolina (USC), and each participant provided written informed consent at enrollment. For further participant characteristics refer to Table 1.

### 2.2. Cognitive Testing and Blood Draw

Cognitive function was assessed using the MoCA Test. The MoCA Test is a short cognitive screening tool that evaluates several cognitive domains, including executive function, visuospatial abilities, short-term memory, attention and working memory, language, concentration, verbal abstraction, and orientation [20]. It has a maximum score of 30, indicating that all questions were answered correctly, with scores below 26 typically indicating a heightened risk of cognitive impairment [20]. HbA1c levels were measured using a nonfasting blood draw [19]. The HbA1c test reflects average blood glucose levels over the preceding three months and is not significantly influenced by the fasting status of the patient on the day of testing [21].

### 2.3. Neuroimaging/Quantifying WMHs

Participants underwent neuroimaging on a Siemens Prisma 3T scanner equipped with a 20-channel head coil in the McCausland Center for Brain Imaging (MCBI) at Prisma Health Hospital, Columbia, SC, as described in detail by Newman-Norlund et al. [11]. A Siemens T2 Fluid-attenuated inversion recovery (FLAIR) MRI sequence was acquired with the following parameters: repetition time (TR) = 5000 ms, echo time (TE) = 387 ms, inversion time (TI) = 1800 ms, variable flip angle, field of view = 230 × 230 × 173 mm, voxel size = 0.9 × 0.9 × 0.9 mm, and GRAPPA acceleration factor = 2 [19]. The severity of WMHs was assessed on FLAIR images by two trained, independent raters using the Fazekas scale [22]. Both raters assessed all images, and any discrepancies in their ratings were resolved through a joint review and discussion until consensus was reached. Ratings were performed separately for deep and periventricular WMHs, with each region assigned a Fazekas score ranging from 0 to 3. A score of 0 indicates no WMHs; 1 indicates small, punctate foci of hyperintensity; 2 reflects confluent patches of hyperintensities separated by areas of normal white matter; and 3 represents the most severe WMHs, characterized by large areas of confluent hyperintensities [22]. The scores for deep and periventricular WMHs were summed to derive the total Fazekas score used in our study, which can range from 0 to 6. Examples of different Fazekas scores are shown in Figure 1.

### 2.4. Statistical Analysis

To test the relationships between HbA1c, total Fazekas scores, and total MoCA scores, we first conducted Spearman’s correlation analyses and regression analyses. Next, mediation analysis was conducted using HbA1c as the independent variable, MoCA total scores as the dependent variable, and Fazekas total scores as the mediating variable. Participant diagnosis of hypertension was controlled for to account for possible confounding effects. Mediation analysis was performed using the Baron and Kenny approach [23]. To determine whether the mediation effect was independent of age, we repeated the analyses while additionally controlling for age.

The indirect effect was calculated by multiplying the coefficient from the regression of the total Fazekas score on HbA1c by the coefficient from the regression of the total MoCA score on the total Fazekas score. Significance testing of the indirect effect was conducted using bootstrapping with 5000 resamples. All analyses were carried out using RStudio (version 2022.11.0-daily+246; Posit Software, PBC, Boston, MA, USA).

## 3. Results

### 3.1. Demographics

We enrolled 205 participants (73.7% female; average age: 46.8 ± 19.9 years, range = 20–79). The average HbA1c was 5.5% (±0.62, range: 4.2–10.8). The average total Fazekas score was 2.55 (±1.63, range: 0–6), and the average MoCA score was 27.24 (±2.56, range: 15–30).

### 3.2. Associations Between Hba1c, WMHs, and Cognition

A negative correlation was observed between HbA1c and MoCA scores (r(205) = −0.408, *p* < 0.001) as well as between the total Fazekas scores and MoCA scores (r(205) = −0.375, *p* < 0.001). Conversely, a positive correlation was found between HbA1c and total Fazekas scores (r(205) = 0.397, *p* < 0.001). Furthermore, age was positively correlated with HbA1c (r(205) = 0.588, *p* < 0.001) and the total Fazekas scores (r(205) = 0.623, *p* < 0.001), and negatively correlated with the MoCA scores (r(205) = −0.379, *p* < 0.001). Age was not included as a covariate in the subsequent mediation analysis due to multicollinearity concerns, as we observed a strong positive correlation between age and HbA1c.

For each 1-point increase in HbA1c there was a 1.53-point decrease in the MoCA score. Similarly, each 1-point increase in HbA1c was associated with a 0.80-point increase in total Fazekas score, and each 1-point increase in total Fazekas score was associated with a 0.54-point decrease in the MoCA score. For each of the above regression analyses, the significance level was *p* < 0.001.

### 3.3. Mediation Analysis

There was a significant direct effect of HbA1c on MoCA total scores, and this effect was partially mediated by the total Fazekas scores. This partial mediation indicates that there was both a statistically significant direct (c′) and indirect (ab) effect (Figure 2). The indirect effect (ab) was −0.224 (95% CI: −0.619 to −0.05; *p* = 0.001). The direct effect (c′) was −1.217 (95% CI: −1.82 to −0.80, *p* < 0.001), and the total effect was −1.441. The proportion of the total effect (c) that was mediated by the Fazekas score was calculated by dividing the indirect effect (ab) by the total effect (c): (−0.224)/(−1.441) = 0.156. This indicates that 15.6% of the total effect of HbA1c on the MoCA scores was mediated by WMHs as assessed by the Fazekas scores (refer to Table 2 for details). In our additional mediation analyses controlling for age, the direct effect of HbA1c on MoCA scores remained significant (c′ = −0.820; 95% CI: −1.207 to −0.334; *p* = 0.002), while the indirect effect via Fazekas scores was no longer statistically significant (ab = −0.029; 95% CI: −0.154 to 0.035; *p* = 0.394).

## 4. Discussion

We demonstrated that WMHs partially mediate the relationship between elevated HbA1c levels and cognitive function, supporting the role of small vessel disease as a pathway through which metabolic dysfunction affects brain health. These findings highlight WMHs as a potential neuroimaging marker for assessing the risk of cognitive decline in individuals with type 2 diabetes or prediabetes.

The results underscore the importance of diabetes prevention and the proper management of HbA1c levels in prediabetic and diabetic patients. Although cognitive decline often becomes clinically evident in older age, its underlying causes typically begin in middle age [24], underscoring the need for early prevention and the control of modifiable risk factors. Proactive HbA1c management, through regular screening, lifestyle modifications, and medication can help mitigate long-term cognitive effects, ultimately benefiting individuals’ quality of life and reducing the burden on healthcare systems, long-term care facilities, and families.

In the mediation analysis adjusting for age, the indirect effect of HbA1c on cognition through WMHs was no longer statistically significant. However, the direct effect of HbA1c remained significant, indicating that metabolic dysfunction may impair cognition via both WMH-dependent and WMH-independent pathways. Importantly, age and WMH burden are biologically interrelated, as age is a well-established risk factor for small vessel disease. While statistically adjusting for age is important to address potential confounding, it may also obscure a true mechanistic relationship between metabolic dysfunction and white matter damage. We interpret this as a limitation of the statistical model rather than evidence against the biological plausibility of our hypothesis. In other words, diabetes does not affect cognition through age; rather, both diabetes and age contribute independently to the development of small vessel disease. Therefore, our conclusion that elevated HbA1c contributes to cognitive decline through small vessel disease remains valid and biologically grounded. Nonetheless, these findings should be interpreted with caution, and future research should aim to disentangle these overlapping and interacting pathways more precisely.

Several limitations must be considered. First, the cross-sectional design limits our ability to infer causal relationships between the variables. Longitudinal studies are needed to establish the temporal sequence of HbA1c elevation, WMH development, and cognitive decline more accurately. Future studies aimed at utilizing serial MRIs, cognitive assessments, and HbA1c measurements would provide a more comprehensive and reliable analysis of long-term effects. Second, the use of the Fazekas scale, while a widely accepted measure of WMH severity, is a semi-quantitative assessment, and a more precise volumetric analysis of WMHs could offer a more detailed understanding of their contribution to cognitive impairment. Third, MoCA is known to be a very general measure of cognitive function, and incorporating domain specific cognitive tests, either in place of or alongside the MoCA used in our study, could provide a more detailed assessment of specific areas of cognitive function and should be considered in future research. A fourth limitation involves the demographic makeup of our sample, which included a disproportionately high number of females, Caucasians, and individuals from middle to high socioeconomic backgrounds compared to the general population. This homogeneity limits the external validity of our findings, and future studies should aim for more representative cohorts to enhance generalizability. Fifth, approximately 18% of participants were diagnosed with hypertension. Although hypertension itself was controlled for, the potential influence of antihypertensive medications was not considered and may have independently affected WMH burden. Finally, the partial mediation observed in our study indicates that other underlying mechanisms, not captured here, likely play a role. The list of other potential mediators is extensive, including impaired insulin signaling in the brain, reduced mitochondrial function, and neuroinflammation driven by increased expression of pro-inflammatory cytokines such as NF-kB [25]. Further research into these mechanisms and other unexplored pathways is essential to fully understand the underlying mechanisms by which elevated HbA1c contributes to decreased cognition.

## 5. Conclusions

Our results reinforce the existing evidence of the negative impact type 2 diabetes can have on cognitive function, specifically through elevated Hba1c levels. This study highlights that HbA1c is a significant factor in cognitive decline, in part through its association with WMHs in the brain. Our findings demonstrate that WMHs are a partial mediator of the relationship between increased HbA1c levels and decreased cognition. This suggests that monitoring and controlling HbA1c levels could potentially decrease the risk of cognitive decline, possibly by limiting WMH accumulation.

## Figures and Tables

**Figure 1 brainsci-15-00692-f001:**
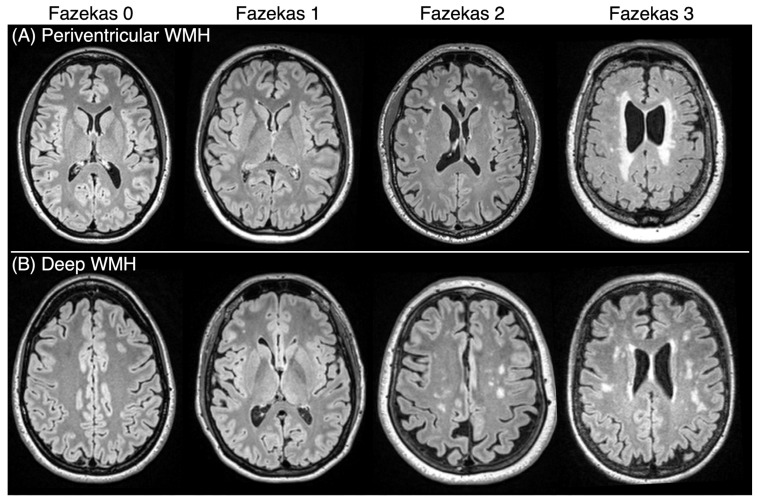
Representative FLAIR images illustrating different Fazekas scores in individual participants. Panel (**A**) displays examples of periventricular white matter hyperintensities (WMHs), while Panel (**B**) highlights examples of deep WMHs. A Fazekas score of 0 indicates the absence of WMHs, whereas a Fazekas score of 3 reflects the most severe form, characterized by confluent WMHs.

**Figure 2 brainsci-15-00692-f002:**
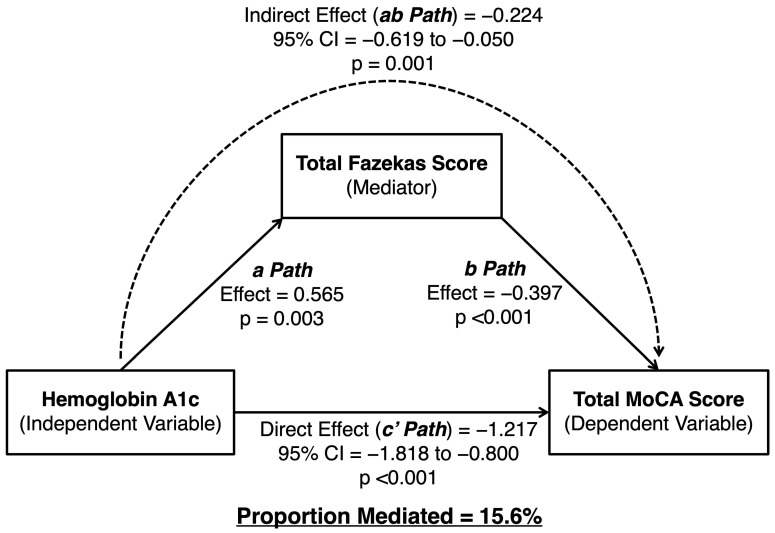
Path diagram showing mediation analysis using HbA1c as the independent variable, Fazekas score as the mediator, and MoCA score as the outcome variable, including hypertension as a control variable. The indirect effect is represented by a dashed arrow.

**Table 1 brainsci-15-00692-t001:** Characteristics of the participants.

Characteristics (*N* = 205)	Mean (SD) or Percentage
Age (years)	46.8 (19.9)
Sex (females:males)	73.7%:25.9%
Race (percentage)	
Asian	4.46%
African American	11.88%
White	83.66%
Hypertension (%) Highest Level of Education (percentage)	18.54%
Completed High School	9.27%
Completed at least one year of college or specialized training	9.27%
Completed College/University	41.95%
Completed Graduate School	37.56%
Other	0.98%
Socioeconomic Status	
High	33.16%
Middle	43.52%
Low	23.32%
HbA1c (%)	5.50 (0.62)
MoCA Total Score	27.24 (2.56)
Fazekas Score Total	2.55 (1.63)
Fazekas Score Deep	1.08 (1.07)
Fazekas Score Periventricular	1.47 (0.76)

SD: standard deviation, HbA1c: hemoglobin A1c, MoCA: Montreal Cognitive Assessment.

**Table 2 brainsci-15-00692-t002:** Results of the mediation analysis controlling for hypertension (N = 205).

	Estimate	Standard Error	t Value	*p* Value
Model 1. Dependent variable: MoCA score				
Model (*R*^2^ = 0.197, adjusted *R*^2^ = 0.185, *p* <0.001)				
**Intercept**	35.243	1.584	22.254	<0.001
**HbA1c**	−1.441	0.292	−4.936	<0.001
**Hypertension**	−0.372	0.467	−0.797	0.427
Model 2. Dependent variable: Fazekas score				
Model (*R*^2^ = 0.136, adjusted *R*^2^ = 0.127, *p* <0.001)				
**Intercept**	−0.736	1.011	−0.728	0.468
**HbA1c**	0.565	0.186	3.029	0.003
**Hypertension**	0.939	0.298	3.147	0.002
Model 3. Dependent variable: MoCA score				
Model (*R*^2^ = 0.197, *R*^2^ = 0.189, *p* < 0.001)				
**Intercept**	34.951	1.538	22.728	<0.001
**Fazekas Score**	−0.397	0.107	−3.713	<0.001
**HbA1c**	−1.217	0.290	−4.204	<0.001
**Hypertension**	0.0003	0.464	0.001	0.999

Abbreviations: MoCA: Montreal Cognitive Assessment, HbA1c: hemoglobin A1c.

## Data Availability

Data supporting the reported results can be requested through our database request form: https://abc.sc.edu/abc-repository-data-requests/. URL accessed on 26 June 2025.

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
