# Peer review of "White Matter Hyperintensities Mediate the Negative Impact of HbA1c Levels on Cognitive Function"

_brainsci, 2025, doi:10.3390/brainsci15070692_

Round 1
Reviewer 1 Report
Comments and Suggestions for Authors
This short report submitted by Johnstone and team, examines the mediating effect of white matter hyperintensities (WMHs) in the association between hemoglobin A1c (HbA1c) levels and cognitive ability. They do so by using data from 205 neurologically healthy adults. Authors show that WMHs, measured through the Fazekas scale, mediate the detrimental effect of high HbA1c on Montreal Cognitive Assessment (MoCA) score to some extent, explaining 15.6% of the overall effect. The research is based on a clear and articulated hypothesis with proper methodology. such as bootstrapped mediation analysis and adjustment for hypertension. The application of a large, robust dataset and validated measures of cognition and imaging enhances the validity of the findings. Nevertheless, although the results support a partial mechanistic association between metabolic dysfunction and cognitive impairment through WMHs, cross-sectional design does not allow causal inference. In addition, the use of the semi-quantitative Fazekas scale rather than volumetric WMH analysis reduces the level of granularity of results.
While the manuscript itself is well-presented & supported by data, I am concerned with the 29% plagiarism reported which is a concern and needs to be remedied to uphold academic integrity. Authors should rewrite or accurately cite all parts indicated for similarity in order to lower the similarity index below suitable thresholds.
Also, authors need to admit more clearly the limitations of causal inference and recommend avenues for longitudinal follow-up studies. This they have missed.
Also please add volumetric WMH analysis or discuss its possible benefits.
Treat multicollinearity more extensively and examine statistical alternatives to control for age.
Elaborate on some possible mediators (e.g., insulin resistance, inflammation) that were not under investigation.
Author Response
Comment 1: This short report submitted by Johnstone and team, examines the mediating effect of white matter hyperintensities (WMHs) in the association between hemoglobin A1c (HbA1c) levels and cognitive ability. They do so by using data from 205 neurologically healthy adults. Authors show that WMHs, measured through the Fazekas scale, mediate the detrimental effect of high HbA1c on Montreal Cognitive Assessment (MoCA) score to some extent, explaining 15.6% of the overall effect. The research is based on a clear and articulated hypothesis with proper methodology. Such as bootstrapped mediation analysis and adjustment for hypertension. The application of a large, robust dataset and validated measures of cognition and imaging enhances the validity of the findings. Nevertheless, although the results support a partial mechanistic association between metabolic dysfunction and cognitive impairment through WMHs, cross-sectional design does not allow causal inference.
Response 1: We appreciate this insight and recognize this as a limitation of our study. Below is a portion of our discussion section that highlights this limitation. The bolded portion has been added to the original submission to help emphasize this limitation and provide guidance for how it could potentially be avoided in future studies.
“Although our study provides insight into the mechanisms through which type 2 diabetes affects cognition, several limitations must be considered. First, the cross-sectional design limits our ability to infer causal relationships between the variables. Longitudinal studies are needed to establish the temporal sequence of HbA1c elevation, WMH development, and cognitive decline more accurately. Future studies aimed at utilizing serial MRIs, cognitive assessments, and HbA1c measurements would provide a more comprehensive and reliable analysis of long-term effects.”
Comment 2: In addition, the use of the semi-quantitative Fazekas scale rather than volumetric WMH analysis reduces the level of granularity of results.
Response 2: We recognize the limitation of using the semi-quantitative Fazekas scale in place of volumetric WMH analysis and will address this point in more detail in a later comment.
Comment 3: While the manuscript itself is well-presented & supported by data, I am concerned with the 29% plagiarism reported which is a concern and needs to be remedied to uphold academic integrity. Authors should rewrite or accurately cite all parts indicated for similarity in order to lower the similarity index below suitable thresholds.
Response 3: Thank you for bringing this to our attention. Academic integrity is of the utmost importance to us. To address this concern, we have carefully reviewed our paper for any claims lacking proper citation and have added the appropriate references where necessary.
Comment 4: Also, authors need to admit more clearly the limitations of causal inference and recommend avenues for longitudinal follow-up studies. This they have missed.
Response 4: Thank you for this note. Below is a portion of our discussion section with the newly added text in bold. We recognize this same section is used to address your other concern regarding the limitations of cross-sectional design and the inability to infer causality. We believe it adequately addresses that concern while also more clearly admitting the limitations and recommending guidance for longitudinal follow-up studies.
“Although our study provides insight into the mechanisms through which type 2 diabetes affects cognition, several limitations must be considered. First, the cross-sectional design limits our ability to infer causal relationships between the variables. Longitudinal studies are needed to establish the temporal sequence of HbA1c elevation, WMH development, and cognitive decline more accurately. Future studies aimed at utilizing serial MRIs, cognitive assessments, and HbA1c measurements would provide a more comprehensive and reliable analysis of long-term effects.”
Comment 5: Also please add volumetric WMH analysis or discuss its possible benefits.
Response 5: Thank you for this note. We agree that the Fazekas scale, being a semi-quantitative measure, offers less granularity than volumetric assessments of WMH. We acknowledged this limitation in our discussion section as follows:
“Second, the use of the Fazekas scale, while a widely accepted measure of WMH severity, is a semi-quantitative assessment, and a more precise volumetric analysis of WMHs could offer a more detailed understanding of their contribution to cognitive impairment.”
To address this concern further, we conducted an additional mediation analysis using the subset of participants for whom WMH volumetric data were available (n = 182). A mediation analysis using this subset yielded results consistent with our original finding, showing that WMH volume partially mediated the relationship between HbA1c and MoCA scores. The proportion mediated was approximately 12.5% and statistically significant at an alpha level of 0.05.
These consistent findings suggest that, despite its lower granularity, the Fazekas scale performs comparably to volumetric WMH measures in this context. Additionally, the Fazekas scale offers greater feasibility for replication in both clinical and large-scale research settings, as it is easier to implement than volumetric WMH extraction. Taken together, our results support the utility and reliability of the Fazekas scale for assessing WMH burden in studies of cognitive aging. For these reasons, and given the larger available sample size with Fazekas ratings, we chose to retain the Fazekas scale as the mediator rather than replacing it with volumetric WMH data.
Comment 6: Treat multicollinearity more extensively and examine statistical alternatives to control for age.
Response 6: We appreciate this suggestion and have conducted both SEM and residualization and found that the direct effect of HbA1c on MoCA remained significant after age adjustment. However, the indirect effect via WMH burden was no longer statistically significant once age was accounted for. This suggests that the mediating role of WMH in the HbA1c–cognition relationship is largely age-dependent. In other words, while WMH burden is a potential pathway through which elevated blood sugar impacts cognition, this effect may reflect an age-related accumulation of small vessel disease, rather than a standalone effect of HbA1c.
We have also added to our methods section, results section, and the following to our discussion section to address and interpret the implication of this finding within the context of our study.
“While WMHs initially appeared to mediate the relationship between HbA1c and cognitive performance, this effect was no longer significant after controlling for age. This suggests that the impact of elevated HbA1c on cognition may be partially explained by age-related increases in WMH, rather than a direct, age-independent mechanism. Importantly, the direct effect of HbA1c on MoCA remained significant even after age adjustment, indicating that metabolic dysfunction may impair cognition via both WMH-dependent and WMH-independent pathways.”
Comment 7: Elaborate on some possible mediators (e.g., insulin resistance, inflammation) that were not under investigation.
Response 7: Thank you for this comment. We have added the following to our discussion section to elaborate on other potential mediators and address the need for further research:
“The list of other potential mediators is extensive, including impaired insulin signaling in the brain, reduced mitochondrial function, and neuroinflammation driven by increased expression of pro-inflammatory cytokines such as NF-kB [24]. Further research into these mechanisms and other unexplored pathways is essential to fully understand the underlying mechanisms by which elevated HbA1c contributes to decreased cognition."
Reviewer 2 Report
Comments and Suggestions for Authors
Comments about the Abstract:
- Lines 19-20, "Type 2 diabetes is linked to impaired cognitive function, but the underlying mechanisms remain poorly understood. Kindly include a brief description of the commonly accepted mechanisms underlying the association between type 2 diabetes and cognitive impairment.
- While the Fazekas scale is a valid and widely used clinical tool, it has limitations in sensitivity and precision due to its ordinal and subjective nature. For mediation analyses and more detailed exploration of WMH-cognition relationships, it would strengthen the study to validate findings using quantitative volumetric MRI measures (e.g., with FSL, SPM, or FreeSurfer).
-
The conclusion statement, "This study suggests that white matter damage represents a critical mechanism linking chronically elevated blood glucose to reduced cognitive function in individuals with impaired glucose metabolism and type 2 diabetes," may overstate the findings. Since your study is observational and based on mediation analysis using the Fazekas scale, it supports an association rather than a causal link. I suggest revising this statement for scientific accuracy. For example:
Suggested Revision:
"This study suggests that white matter hyperintensities may partially mediate the association between chronically elevated blood glucose levels and cognitive impairment in individuals with impaired glucose metabolism and type 2 diabetes, supporting a potential microvascular mechanism."
Comments about the Introduction:
- Lines 34-35: The relationship between chronically elevated blood sugar, as seen in type 2 diabetes, and impaired cognitive function is well established..... Please add a citation to this statement.
- Line 43: Please replace "insulin resistance in the brain" with cerebral insulin resistance.
- Lines 53-54: Please specify the metabolic conditions and cardiovascular diseases for better understanding.
Comments about the Materials and Methods:
- Please move Table 1 under the Participants section.
- To improve measurement precision and enhance the rigor of the mediation analysis, i recommend incorporating volumetric WMH quantification using automated tools such as FSL (BIANCA), FreeSurfer, or SPM.
- This study should employ longitudinal designs to establish the temporal sequence and better infer causality in the HbA1c , WMHs, cognition pathway.
- Please include age as a covariate using residualization techniques or structural equation modeling (SEM). This will help to isolate the independent effects of HbA1c and aging on WMH burden and cognition.
- It is suggested that MoCA be supplemented with domain-specific neuropsychological tests (e.g., Trail Making Test for executive function, Digit Span for working memory) because MoCA is relatively coarse and does not capture domain-specific deficits.
Summary of review report:
- This study addresses an important research question and presents statistically significant findings. However, its scientific rigor is limited by certain methodological choices, including the use of a semi-quantitative measure for WMH burden, a cross-sectional study design, and insufficient control for key confounding variables such as age. As a result, the Results, Discussion, and Conclusion sections are relatively brief and lack the depth needed to fully engage readers or encourage future research building upon these findings.
Author Response
Comment 1: Lines 19-20, "Type 2 diabetes is linked to impaired cognitive function, but the underlying mechanisms remain poorly understood. Kindly include a brief description of the commonly accepted mechanisms underlying the association between type 2 diabetes and cognitive impairment.
Response 1: We appreciate this suggestion and have adjusted the following paragraph in our Introduction to address it. The bolded text represents additions to our original draft providing further detail on the potential mechanisms underlying the association between type 2 diabetes and cognitive impairment.
“Several mechanisms have been proposed to explain the link between diabetes and cognitive decline, including cerebral insulin resistance, chronic inflammation, and microangiopathy [2], [3]. Chronic hyperglycemia contributes to each of these processes, ultimately causing oxidative stress and neuronal injury or apoptosis, which likely underlie the resulting cognitive dysfunction (Gupta et al). Both inflammation and microangiopathy also significantly contribute to the development of white matter hyperintensities (WMHs) [4], areas of increased signal intensity on T2-weighted MRI scans that reflect white matter and cerebral small vessel damage [5]. In type 2 diabetes, factors such as endothelial cell dysfunction, oxidative stress, advanced glycation end products, and inflammation may be directly responsible for this damage [1]. Specifically, the thickening and injury of small blood vessels can lead to decreased blood flow to critical areas of the brain, potentially resulting in the formation of WMHs and contributing to cognitive impairment [6].”
Comment 2: While the Fazekas scale is a valid and widely used clinical tool, it has limitations in sensitivity and precision due to its ordinal and subjective nature. For mediation analyses and more detailed exploration of WMH-cognition relationships, it would strengthen the study to validate findings using quantitative volumetric MRI measures (e.g., with FSL, SPM, or FreeSurfer).
Response 2: Thank you for this thoughtful suggestion. As this concern was also raised by another reviewer, we conducted a follow-up mediation analysis using volumetric WMH data. We also previously acknowledged in our discussion that the Fazekas scale, while widely used, is less granular than volumetric methods:
“Second, the use of the Fazekas scale, while a widely accepted measure of WMH severity, is a semi-quantitative assessment, and a more precise volumetric analysis of WMHs could offer a more detailed understanding of their contribution to cognitive impairment.”
To validate our findings, we re-ran the mediation analysis in the subset of 182 participants for whom WMH volumes were available. This analysis yielded results consistent with our original findings: WMH volume significantly and partially mediated the relationship between HbA1c and MoCA scores, with a mediated proportion of approximately 12.5% (p < 0.05).
These consistent results suggest that the Fazekas scale performs comparably to volumetric WMH measures as a mediator in this context. Furthermore, the use of Fazekas allowed us to include a larger sample and enhances the feasibility of replication in clinical and large-scale studies, given its accessibility and ease of implementation. For these reasons, we chose to retain the Fazekas scale as the mediator rather than replacing it with volumetric WMH data.
Comment 3: The conclusion statement, "This study suggests that white matter damage represents a critical mechanism linking chronically elevated blood glucose to reduced cognitive function in individuals with impaired glucose metabolism and type 2 diabetes," may overstate the findings. Since your study is observational and based on mediation analysis using the Fazekas scale, it supports an association rather than a causal link. I suggest revising this statement for scientific accuracy. For example:
Suggested Revision:
"This study suggests that white matter hyperintensities may partially mediate the association between chronically elevated blood glucose levels and cognitive impairment in individuals with impaired glucose metabolism and type 2 diabetes, supporting a potential microvascular mechanism."
Response 3: We believe this is an excellent suggestion that more appropriately conveys the overall findings from our study. We appreciate the thoughtful suggestion and have replaced our original conclusion statement with the suggested revision.
Comment 4: Comments about the Introduction: Lines 34-35: The relationship between chronically elevated blood sugar, as seen in type 2 diabetes, and impaired cognitive function is well established..... Please add a citation to this statement.
Response 4: Thank you for noting this. Proper citations have been added that support this statement. Both the papers used for the citation can be found below:
Sebastian MJ, Khan SK, Pappachan JM, Jeeyavudeen MS. Diabetes and cognitive function: An evidence-based current perspective. World J Diabetes. 2023;14(2):92-109. doi:10.4239/wjd.v14.i2.92
Zheng F, Yan L, Yang Z, Zhong B, Xie W. HbA1c, diabetes and cognitive decline: the English Longitudinal Study of Ageing. Diabetologia. 2018;61(4):839-848. doi:10.1007/s00125-017-4541-7
Comment 5: Line 43: Please replace "insulin resistance in the brain" with cerebral insulin resistance.
Response 5: We appreciate this suggestion and have changed the wording from “insulin resistance in the brain” to “cerebral insulin resistance”.
Comment 6: Lines 53-54: Please specify the metabolic conditions and cardiovascular diseases for better understanding.
Response 6: We agree that such specifications would make for a clearer and better understanding. Specific metabolic conditions and cardiovascular diseases such as hypertension, carotid atherosclerosis, atrial fibrillation, and diabetes have been specified here and an additional citation has been made to further back up the claims that white matter hyperintensities are a significant factor in cognitive impairment, particularly in individuals with certain metabolic conditions and cardiovascular diseases. The modified statement can be found below:
“Additionally, growing evidence identifies WMHs as a significant factor in cognitive impairment, particularly in individuals with certain metabolic conditions and cardiovascular diseases such as hypertension, carotid atherosclerosis, atrial fibrillation, and diabetes [14], [15].”
Comment 7: Comments about the Materials and Methods: Please move Table 1 under the Participants section.
Response 7: Thank you for this suggestion. Table 1 has been moved under the Participants section.
Comment 8: To improve measurement precision and enhance the rigor of the mediation analysis, i recommend incorporating volumetric WMH quantification using automated tools such as FSL (BIANCA), FreeSurfer, or SPM.
Response 8: Thank you for this comment. We appreciate your emphasis on the importance of volumetric WMH quantification. As this point was raised previously, we refer you to our earlier response addressing the limitations of the Fazekas scale and describing our additional mediation analysis using WMH volume data in a subset of participants. In brief, that analysis yielded results consistent with our original findings, suggesting that the Fazekas scale performs comparably to volumetric measures in this context, while also allowing for a larger sample size and greater feasibility in broader research and clinical settings.
Comment 9: This study should employ longitudinal designs to establish the temporal sequence and better infer causality in the HbA1c , WMHs, cognition pathway.
Response 9: Thank you for this suggestion. We agree that a longitudinal design would better support causality in the relationship between HbA1c, WMHs, and cognition. However, our study was limited to cross-sectional data. This limitation is addressed in the following portion of our discussion section, which acknowledges the constraint and highlights the opportunity for future studies to conduct similar analyses using a more longitudinal design.
“Although our study provides insight into the mechanisms through which type 2 diabetes affects cognition, several limitations must be considered. First, the cross-sectional design limits our ability to infer causal relationships between the variables. Longitudinal studies are needed to establish the temporal sequence of HbA1c elevation, WMH development, and cognitive decline more accurately. Future studies aimed at utilizing serial MRIs, cognitive assessments, and HbA1c measurements would provide a more comprehensive and reliable analysis of long-term effects.”
Comment 10: Please include age as a covariate using residualization techniques or structural equation modeling (SEM). This will help to isolate the independent effects of HbA1c and aging on WMH burden and cognition.
Response 10: We appreciate this suggestion and have conducted both SEM and residualization and found that the direct effect of HbA1c on MoCA remained significant after age adjustment. However, the indirect effect via WMH burden was no longer statistically significant once age was accounted for. This suggests that the mediating role of WMH in the HbA1c–cognition relationship is largely age-dependent. In other words, while WMH burden is a potential pathway through which elevated blood sugar impacts cognition, this effect may reflect an age-related accumulation of small vessel disease, rather than a standalone effect of HbA1c.
We have also added to our methods section, results section, and the following to our discussion section to address and interpret the implication of this finding within the context of our study.
“In the mediation analysis adjusting for age, the indirect effect of HbA1c on cognition through WMHs was no longer statistically significant. However, the direct effect of HbA1c remained significant, indicating that metabolic dysfunction may impair cognition via both WMH-dependent and WMH-independent pathways. Importantly, age and WMH burden are biologically interrelated, as age is a well-established risk factor for small vessel disease. While statistically adjusting for age is important to address potential confounding, it may also obscure true mechanistic relationship between metabolic dysfunction and white matter damage. We interpret this as a limitation of the statistical model rather than evidence against the biological plausibility of our hypothesis. In other words, diabetes does not affect cognition through age; rather, both diabetes and age contribute independently to the development of small vessel disease. Therefore, our conclusion, that elevated HbA1c contributes to cognitive decline through small vessel disease, remains valid and biologically grounded. Nonetheless, these findings should be interpreted with caution, and future research should aim to disentangle these overlapping and interacting pathways more precisely."
Comment 11: It is suggested that MoCA be supplemented with domain-specific neuropsychological tests (e.g., Trail Making Test for executive function, Digit Span for working memory) because MoCA is relatively coarse and does not capture domain-specific deficits.
Response 11: We appreciate this suggestion and fully agree that incorporating neuropsychological tests targeting specific cognitive domains, such as the Trail Making Test for executive function or the Digit Span test for working memory, would allow for a more nuanced understanding of cognitive performance. Unfortunately, these measures were not available in our dataset and could therefore not be included in the present analysis. We have added the following sentence to our discussion to acknowledge this limitation and to encourage future research to include such domain-specific assessments:
“Furthermore, MoCA is known to be a very general measure of cognitive function and incorporating domain specific cognitive, either in place of or alongside MoCA used in our study, could provide a more detailed assessment of specific areas of cognitive function and should be considered in future research.
Comment 12: Summary of review report: This study addresses an important research question and presents statistically significant findings. However, its scientific rigor is limited by certain methodological choices, including the use of a semi-quantitative measure for WMH burden, a cross-sectional study design, and insufficient control for key confounding variables such as age. As a result, the Results, Discussion, and Conclusion sections are relatively brief and lack the depth needed to fully engage readers or encourage future research building upon these findings.
Response 12: Thank you for all your constructive comments and suggestions. We hope the additions and edits that have been made to our paper as well as the provided rationale for our methodological choices help address your concerns and strengthen the overall impact of our work.
Reviewer 3 Report
Comments and Suggestions for Authors
There are several studies demonstrating a role of HbA1c in cognitive decline, as well as with white matter hyperintensity (WMH) (see, for example, https://pubmed.ncbi.nlm.nih.gov/29368156/ https://pmc.ncbi.nlm.nih.gov/articles/PMC10958578/).
Here, on a small cohort of neurologically intact patients, the authors demonstrated that WMHs partially mediate the relationship between elevated levels of HbA1c and cognitive decline.
Please find my comments below.
In the introduction, please briefly explain what HbA1c is.
The cohort predominantly comprised white females, primarily from middle or high socioeconomic backgrounds. What if you performed analysis in sub-groups of the cohort (males only, African Americans only, participants with low socioeconomic status only, etc), would the results be the same?
Presumably, some of the study participants, might have already taken antidiabetic medications (or at least antihypertensive medications, as about 18 percent of participants had hypertension; notably, hypertension itself is a major risk factor for cognitive impairment, and treating hypertension can help reducing such risk; furthermore, antihypertensive medications might affect WMHs). This was not taken into consideration and should be at least discussed.
You state that “Finally, the partial mediation observed in our study indicates that other underlying mechanisms, not captured here, likely play a role and should be explored in future research”. Can you add couple of sentences about possible other mechanisms?
Author Response
Comment 1: There are several studies demonstrating a role of HbA1c in cognitive decline, as well as with white matter hyperintensity (WMH) (see, for example, https://pubmed.ncbi.nlm.nih.gov/29368156/ https://pmc.ncbi.nlm.nih.gov/articles/PMC10958578/). Here, on a small cohort of neurologically intact patients, the authors demonstrated that WMHs partially mediate the relationship between elevated levels of HbA1c and cognitive decline.
Response 1: Thank you for mentioning these relevant references. We agree that there is substantial existing literature demonstrating associations between elevated HbA1c, cognitive decline, and WMH burden. These references further support the rationale for our investigation, and we have now included them to strengthen the background of our manuscript. However, to our knowledge, prior studies have not specifically examined WMHs as a mediator in the relationship between HbA1c and cognitive function, which we believe is a novel contribution of our work.
Comment 2: In the introduction, please briefly explain what HbA1c is.
Response 2: Thank you for this note. It is important to us that our readers understand what HbA1c measures, highlighting why it was used in our study and other studies that are referenced within this paper. The following has been added to our introduction to better explain what HbA1c is.
“Hemoglobin A1c (HbA1c) is a measure of glycated hemoglobin that reflects average blood glucose levels over the previous 2 to 3 months, with levels over 6.5% being diagnostic for diabetes [Sherwani et al.].”
Comment 3: The cohort predominantly comprised white females, primarily from middle or high socioeconomic backgrounds. What if you performed analysis in sub-groups of the cohort (males only, African Americans only, participants with low socioeconomic status only, etc), would the results be the same?
Response 3: We appreciate this suggestion and agree that this is a limitation of our study. Our major concern with analyzing individual subgroups is that the sample groups would not be large enough for the results to be statistically meaningful. We recognize this is partially due to our sample not being proportionally representative of the general population and have now directly addressed this limitation in the discussion section of our paper with the following statements.
“A third limitation involves the demographic makeup of our sample, which included a disproportionately high number of females, Caucasians, and individuals from middle to high socioeconomic backgrounds compared to the general population. This homogeneity limits the external validity of our findings, and future studies should aim for more representative cohorts to enhance generalizability.”
Comment 4: Presumably, some of the study participants, might have already taken antidiabetic medications (or at least antihypertensive medications, as about 18 percent of participants had hypertension; notably, hypertension itself is a major risk factor for cognitive impairment, and treating hypertension can help reducing such risk; furthermore, antihypertensive medications might affect WMHs). This was not taken into consideration and should be at least discussed.
Response 4: We appreciate you bringing these points to our attention and believe it is important to acknowledge potential factors that may influence WMH burden.
Regarding antidiabetic medications, we chose HbA1c as our primary measure of glycemic control because it reflects the cumulative effects of both disease severity and treatment efficacy. As such, we believe adjusting for diabetes medications is not necessary, since their impact would be reflected in HbA1c levels themselves.
With respect to hypertension, we did adjust for its presence in our analyses. However, we acknowledge that the use of antihypertensive medications may have independent effects on WMH burden that were not accounted for. We have now added the following sentence to our discussion to address this limitation:
“Fourth, approximately 18% of participants were diagnosed with hypertension. Although hypertension itself was controlled for, the potential influence of antihypertensive medications was not considered and may have independently affected WMH burden.”
We appreciate your suggestion and believe this addition strengthens the transparency of our study.
Comment 5: You state that “Finally, the partial mediation observed in our study indicates that other underlying mechanisms, not captured here, likely play a role and should be explored in future research”. Can you add couple of sentences about possible other mechanisms?
Response 5: Thank you for this suggestion. We have added the following to our discussion section to address this point:
“Finally, the partial mediation observed in our study indicates that other underlying mechanisms, not captured here, likely play a role. The list of other potential mediators is extensive, including impaired insulin signaling in the brain, reduced mitochondrial function, and neuroinflammation driven by increased expression of pro-inflammatory cytokines such as NF-kB [Zilliox et al.]. Further research into these mechanisms, the mechanism explored in this paper, as well as other that have not yet been explored, is crucial for understanding the underlying mechanisms by which elevated HbA1c contributes to decreased cognition.”
We hope this addition, along with the discussion of potential mechanisms in our introduction, sufficiently addresses other possibilities and helps to highlight directions for future research. While we recognize that we do not propose any mechanisms beyond those previously reported in literature, we are hesitant to speculate on mechanisms that fall outside the scope of our study or existing research.
Round 2
Reviewer 1 Report
Comments and Suggestions for Authors
Authors have revised the article successfully.
Reviewer 2 Report
Comments and Suggestions for Authors
The authors have incorporated the suggested revisions; therefore, it is recommended that the manuscript be accepted for publication